# 3D Nanoprinting of All-Metal Nanoprobes for Electric AFM Modes

**DOI:** 10.3390/nano12244477

**Published:** 2022-12-17

**Authors:** Lukas Matthias Seewald, Jürgen Sattelkow, Michele Brugger-Hatzl, Gerald Kothleitner, Hajo Frerichs, Christian Schwalb, Stefan Hummel, Harald Plank

**Affiliations:** 1Christian Doppler Laboratory for Direct-Write Fabrication of 3D Nano-Probes, Graz University of Technology, 8010 Graz, Austria; 2Institute of Electron Microscopy and Nanoanalysis, Graz University of Technology, 8010 Graz, Austria; 3Graz Centre for Electron Microscopy, Steyrergasse 17, 8010 Graz, Austria; 4GETec Microscopy Inc., 1020 Wien, Austria; 5Quantum Design Microscopy, 64293 Darmstadt, Germany

**Keywords:** 3D nanoprinting, additive manufacturing, direct-write nanofabrication, focused electron beam induced deposition, metal nanostructures, platinum, atomic force microscopy, conductive atomic force microscopy, electrostatic force microscopy

## Abstract

3D nanoprinting via focused electron beam induced deposition (FEBID) is applied for fabrication of all-metal nanoprobes for atomic force microscopy (AFM)-based electrical operation modes. The 3D tip concept is based on a hollow-cone (HC) design, with all-metal material properties and apex radii in the sub-10 nm regime to allow for high-resolution imaging during morphological imaging, conductive AFM (CAFM) and electrostatic force microscopy (EFM). The study starts with design aspects to motivate the proposed HC architecture, followed by detailed fabrication characterization to identify and optimize FEBID process parameters. To arrive at desired material properties, e-beam assisted purification in low-pressure water atmospheres was applied at room temperature, which enabled the removal of carbon impurities from as-deposited structures. The microstructure of final HCs was analyzed via scanning transmission electron microscopy—high-angle annular dark field (STEM-HAADF), whereas electrical and mechanical properties were investigated in situ using micromanipulators. Finally, AFM/EFM/CAFM measurements were performed in comparison to non-functional, high-resolution tips and commercially available electric probes. In essence, we demonstrate that the proposed all-metal HCs provide the resolution capabilities of the former, with the electric conductivity of the latter onboard, combining both assets in one design.

## 1. Introduction

Atomic force microscopy (AFM) is a powerful technology applied in many fields of research, ranging from classical material sciences of soft matter to life sciences, to name a few. Aside from imaging a surface’s topography with nanometer and even sub-nanometer resolution in lateral and vertical directions, respectively, AFM allows for simultaneous collection of a variety of physical surface characteristics, such as electric, magnetic, mechanical or thermal properties [1]. However, to access the latter, inherently functionalized AFM probes are required. The most often used approach to provide these functionalities is thin-film coating of prefinished standard nanoprobes, allowing the deposition of a wide range of materials. However, an inherent drawback of such coatings is the unavoidable increase in apex radius, which mostly reduces lateral resolution capabilities [2]. Furthermore, the AFM tip can experience wear effects during measurement, which entails the risk of delamination, as reported in several previous studies [3,4]. This unknown variable reduces the measurement reliability and often acts as limiting factor, especially when working at the lowest nanoscale. An alternative approach to circumvent these challenges is the use of nanoprobes, which are entirely made from materials with desired functionalities [5]. Based on this motivation, here, we present the development of such a concept for application with electrical AFM modes, including conductive AFM (CAFM), electrostatic force microscopy (EFM) or Kelvin force microscopy (KFM). To that end, a high-resolution nanoprobe in the form of a hollow cone (HC) is directly written in the tip region of a cantilever via 3D focused electron beam induced deposition (3D-FEBID). This 3D nanoprinting technology (3DNP) is capable of precisely depositing complex free-standing 3D geometries [6] with nanoscale features from a gaseous precursor with low demands on the substrates [7], which makes this additive manufacturing method an ideal candidate for the modification of prestructured devices such as AFM cantilevers [5,8]. For the application as CAFM probes, a set of requirements has to be fulfilled: (1) a sharp apex, as the measurement resolution strongly depends on the end radius of the probe; (2) high resistance against wear effects in order to maintain lateral resolution and ensure a long lifetime of the tip; and (3) sufficient electrical conductivity, which is addressed in the purification section. A special but highly welcome request is tip fabrication directly on the flat electrode of the cantilever chip (4). Whereas this significantly simplifies the manufacturing of the cantilever platform on one hand, the 3D-FEBID tip must have a minimum size in order to be the lowest point of the AFM setup on the other hand. For such a tall probe geometry (at least 3 µm in height), further important considerations include (5) mechanical stability under force load and (6) deposition times. Among these requirements, the electrical conductivity (aspect 3) is the most critical because, typically for organometallic FEBID precursors, the material consists of a nanogranular microstructure with crystalline metal grains embedded in a carbonaceous matrix [9,10,11,12]. In this state, the electric conductivity is governed by hopping processes in the correlated-variable range-hopping regime and is therefore orders of magnitude lower compared to bulk metals [13]. Furthermore, such materials have a low Young’s modulus [14] and weak resistance against mechanical wear [8], both of which are highly undesirable characteristics for AFM-related applications. However, during the last decade, different in situ and/or post-growth procedures have been demonstrated, enabling controlled functionality tuning, depending on the FEBID precursor used, ranging from thermal annealing [15,16] or co-flow of reactive gases [17,18] over laser exposure [19] to operation in different atmospheres [20], among others [21]. For the Pt-based precursor used in this study, first purification attempts started with O_2_ as annealing gas at variable temperatures, which, aside from very high purities, revealed, that the key requirement is to reduce the gas temperatures to prevent the formation of cracks and/or pores [22]. Based on these findings, the gas was changed to room temperature H_2_O due to higher diffusion constants, which achieved excellent results [23], efficiently removing the carbon [24,25] without the formation of cracks or voids under optimized conditions [12,23,26]. In another study, a comprehensive simulation was set up to explain experimental results and shed light on the dynamics involved [27]. The current model starts with a H_2_O diffusion into the carbon-dominated material, where it generates highly reactive fragments, such as methane, carbon monoxide, etc., upon interaction with the electron beam [24]. Those volatile products diffuse outwards and are pumped away by the vacuum system. In analogy to the deposition process, an efficient purification requires a balance between accessible water molecules within the deposited material and a sufficient number of electrons. Although the carbon removal is accompanied by a high volume loss of approximately 2/3, this comparably gentle treatment can also be applied to fragile 3D geometries [26] and to sensitive substrates, such as the prefinished AFM cantilever used herein. In this work, we elaborate an electrical AFM nanoprobe in four major steps: (1) design and parameter optimization for the initial fabrication of a PtC_X_ hollow cone via 3D-FEBID; (2) adaption of the purification procedure to reliably transfer as-deposited materials into highly conductive platinum structures; (3) comprehensive characterization of mechanical, structural, chemical and electrical properties; and (4) AFM studies with fully optimized HCs, which reveal superior lateral resolution in topography imaging, followed by the demonstration of EFM and CAFM measurements on various materials. This study is motivated by the potential of 3D-FEBIDs to (1) succeed in the fabrication of all-metal 3D nanoprobes for advanced electric AFM measurements with yet unachieved performances and (2) to produce such 3D nanoprobes on challenging surfaces, such as a prefinished AFM cantilever, where alternative approaches cannot be applied. As will be shown, this study proves both aspects, demonstrating the readiness of this technology for industrially relevant applications.

## 2. Methods

3D Nanofabrication—FEBID deposition was performed in a Quanta 3D-FEG SEM/FIB (scanning electron microscope/focused ion beam) dual-beam system (FEI, Eindhoven, The Netherlands). All structures reported in this work were deposited from a platinum precursor (MeCpPt^(IV)^Me_3_, CAS: 94442-22-5). The FEI standard gas injection system (GIS) was installed at an inclination angle of 52° with a distance of 100 μm to the substrate surface and a projected radial distance of 125 μm to the deposition center. The precursor was heated to 45 °C for at least 3 h before each deposition experiment, and the gas flow was opened for at least 5 min prior to every deposition process to establish equilibrium conditions. The chamber pressure was thereby increased from p_0_ ≈ 1 × 10^−6^ mbar to p_d_ ≈ 8 × 10^−6^ mbar during gas injection. SEM images were usually taken from the top or with a tilt angle of 52°, unless otherwise stated. Further analysis was performed with ImageJ^©^, Origin^©^ and Python^©^. Complementary Monte Carlo simulations were carried out with the Casino^©^ package (Version 2.5.1.0).

Purification—Material transfer procedures were performed using the same Quanta 3D FEG instrument (FEI Company, The Netherlands), which allows for low-vacuum operation (20–130 Pa) without breaking the vacuum. The internal patterning engine was used with the same exposure file format as for the initial FEBID fabrication (stream files). Applied parameters are specified in Section 3.2 of the main text. Energy-dispersive X-ray spectroscopy (EDXS) was performed in situ with an EDAX XL-30 system (EDAX Corporation, Mahwah, NJ, USA) at a beam energy of 30 keV and a beam currents of 26 nA and 62 nA for 60 s unless otherwise stated. For EDX analyses, the previously established procedures were followed by means of peak area integration of the C-K_α_ peak in the range of 180–360 eV and in the range of 9.26–9.54 keV for the Pt-L_α_ peak. The Pt-M_α_ peak at 2.05 keV was not used in this study, as the experiments were performed on gold-coated silicon substrates, with the Au-M_α_ peak at 2.12 keV, leading to unwanted but unavoidable overlap with the Pt-M_α_ peak. In previous studies, [23,25,28] the ratio of carbon and platinum peak areas was used as a semiquantitative measure for the chemical evolution to quantify at low intensities in combination with the overlapping peaks; however, it is prone to errors and therefore unreliable.

Transmission Electron Microscopy—Transmission electron microscopy (TEM) and scanning transmission electron microscopy (STEM) measurements were performed with a Tecnai F20 (FEI, the Netherlands) operated at 200 keV under monochromated conditions using bright-field (BF) and high-angle annular dark-field (HAADF) imaging. For TEM studies, the relevant 3D structures were directly fabricated on Cu slit grids (2 mm in diameter) and split in half to provide access to the central grid bar to prevent any further preparation/transfer steps. Analyses were performed using the Digital Micrograph software package (Gatan Microscopy Suite Version 3.30.2016.0).

Atomic Force Microscopy—Atomic force microscopy (AFM) measurements were performed on (1) a FastScanBio AFM (Bruker Corporation, Billerica, MA, USA) operated by a Nanoscope V Controller and (2) on an AFSEM™ (GETec Microscopy, Vienna, Austria) operated by an Anfatec SPM Controller (Anfatec Instruments AG, Oelsnitz, Germany). Conventional AFM for topography scans was performed in tapping mode in air under soft repulsive conditions. Electrostatic force microscopy (EFM) was performed by using the two-pass method, with topography (1st pass) and electrostatic interactions (2nd pass) measured consecutively. The first pass w performed as described for conventional AFM, whereas for the second pass, the AFM cantilever was lifted to a so-called lift height to avoid van der Waals interactions [29]. The lift height has to be adjusted for samples individually to enable stable operation, typically in the range of 10–30 nm, depending on the main scan process parameter. Conductive AFM was performed in contact mode using an AFSEM™ scanner together with built-in electronics for I/U curves. AFM data were post-processed and analyzed with NanoScope Analysis 1.80 and Gwyddion (version 2.61) [30].

## 3. Hollow-Cone Fabrication

Based on a classification by Winkler [31], parameters for 3D-FEBID deposition can be divided into electron-beam setup, patterning strategy and GIS setup. It is worth mentioning that this differentiation cannot omit the fact that the individual parameters do affect the resulting deposit in a convoluted manner [31,32]. Consequently, the identification of successful deposition parameters requires parameter mapping and careful optimization for fabrication of a targeted structure. Because changing the electron-beam setup during deposition is not feasible, as it requires manual refocusing of the beam after changing parameters, the initial focus of exploration was the apex radius, which is highly relevant for high-resolution nanoprobes (requirement 1). For vertical pillars, a 30 keV beam of energy results in the sharpest apices; for lower beam energies, tip shapes are blunter, not fulfilling the highest-priority requirement [5]. Consequently, a beam energy of 30 keV was used for all HC deposition procedures. In this work, the prominent FEBID platinum precursor MeCpPt^(IV)^Me_3_ (CAS: 94442−22−5) was used [31].

### 3.1. Hollow-Cone Deposition

A cone can be geometrically defined by its base diameter and its total height. Whereas the former can be directly addressed via 3DNP, the latter is the result of a complex interplay of beam setup and patterning parameters, which impact the slope angle and therefore require a closer look. To form a true hollow cone (HC), a sequence of discrete concentric circles is used, starting with an outer radius (*r_O_*), which gradually decreases to a minimum inner radius (*r_I_*), as depicted in Figure 1a. To prove the hollow nature of such cones, the patterning was changed to produce half-HCs only, as shown in Figure 1b. For completeness, a fully closed HC was opened via focused ion beam (FIB) processing, revealing the same result, with a hollow inner nature. Within each circle, the e-beam jumps discretely from point to point, with an averaged patterning velocity of v_P_ = PoP/DT, where DT is the local dwell time (green points in (a)), and PoP is the pixel-to-pixel point pitch (indicated in orange). After completing an individual circle, the diameter is reduced by a certain increment, which can be understood as PoP. To distinguish between the two values, the PoP *along circles* is further denoted as *fast-pitch* fp, whereas PoPs *between circles* are denoted as *slow-pitch* sp (see Figure 1a). Although all parameters have coupled cross-influences and this study was conducted according to a multimatrix design, here, we present the most relevant aspects individually, starting with the importance of subsequent circle overlaps controlled by the slow-pitch *sp*. Figure 1d shows three tilted SEM images of sequential, truncated ring structures with sp values of 0.1 nm (top), 1 nm (center) and 2 nm (bottom), whereas all other parameters were kept the same. As evident and basically in agreement with previous findings [5,33], increasing the sp by only 1 nm can lead to a situation in which subsequent volumetric elements start to miss the former ring as a consequence of unstable liftoff behavior. Reducing sp values below 1 nm was found to result in much steeper slopes, requiring very short DTs for compensation and an excessive number of patterning points. The most efficient, predictable and reliable 3D growth was found for sp = 1 nm, in agreement with previous findings; therefore, this value was used for further experiments. The next parameter of exploration is the beam current (I_0_), which was varied from 110 pA to 850 pA, whereas DTs and sp’s were kept the same. However, owing to larger beam diameters for higher currents, absolute fp values were gradually increased but held constant at beam overlaps of about negative 250%. 

Figure 1d shows total HC heights as a function of total exposure times (TET) for different beam currents but identical DTs and sp values, whereas fp values were adopted, enabling constant beam overlaps of negative 250%, which immediately revealed decreasing heights for higher I_0_ values (black–red–blue). This can be rationalized by the lower number of patterning points due increasing fp values, which means reduced overall doses. More importantly, the achievable h_T_’s stagnate for the highest beam current of 850 pA (blue–green), which suggests a strongly depleted precursor situation in the molecule-limited regime, as expected. The inset shows the required TETs for a target height of around 4 μm, where the saturation for beam currents >430 pA becomes clearly evident, which is why the same current was used for further experiments.

The next main parameter to be discussed is the fp within individual circles. Figure 2 shows a series of tilted (upper row) and top-view (bottom row) SEM images, which immediately reveal two facts: first, larger fpvalues (towards the left) lead to the expected formation of pillar-like features, as the consecutive overlap is above the critical value for the formation of smooth surfaces [33]. Second, an azimuthal asymmetry is evident, in agreement with the gas flux indicated by a red arrow. Even for smallest fp values of 42 nm, which appear widely smooth in the titled SEM inspection (upper row), the formation of individual features in the geometrical shadow room is still evident (yellow arrows) [34,35]. To suppress the distinct formation of pillar-like features, the starting point within each single circle was shifted by ±50 % of the fp, effectively rotating the entire ring pattern in such a way that all new pixel points were in between patterning pixels of the previous ring. Despite the simple execution, the effect is remarkable, as shown by the green-framed images on the right in Figure 2, with an extremely smooth surface from the top and from the side. Given these promising results, fpvalues were not further decreased, as growth efficiencies were found to decay due to stronger proximal depletion, in agreement with previous results [33]. Therefore, for further experiments, we always used sp values of 1 nm and fp values of 42 nm for I_0_ of 430 pA, equal to a beam overlap of 250% as a more generic, machine-independent orientation number for replication.

Next, we focus on h_T_ control, which essentially depends on the side wall angles (α) for a given base diameter (d_B_). In principle, there are two main routes, which allow coarse and fine tuning. The first approach involves increasing the number of loops for the same diameter (see grey dashed lines in Figure 1a) while still using the pixel rotation approach discussed above. Secondly, the DT can be adopted accordingly, as discussed below. 

Figure 3a shows an SEM side-view collage of single-loop (l = 1) cones for 200 pA (red), 430 pA (blue) and 850 pA (green), clearly showing the suitability of even higher currents for precise HC fabrication. When increasing the loop number (l) from 1 to 4, h_T_ can strongly but only discretely be increased, as clearly evidenced by the in-scale SEM images. A quantitative summary is shown in (b) for different currents (I_0_), where solid lines represent the more relevant slope angle (α, left ordinate), and dashed lines indicate the more practical heights (h_T_) related to the right ordinate. As evident, this approach allows for height tuning between 2 μm and 7 μm (shaded blue) but with discrete incremental steps of more than 1 μm. α is measured as an average side wall tilt, whereas real slopes become less steep with increased heights. This effect can be seen in the SEM images (a) around the apex region and is attributed to the fact that as the rings and the involved active volumes become smaller, local beam heating increases [12,36,37,38]. In turn, local desorption rates increase, which leads to decreasing volume growth rates (VGR) and slightly smaller angles towards the apex region. This becomes even stronger for taller HCs, as heat removal is complicated by low thermal conductivities, which is evidenced by increasing loop numbers in (a). The saturating α behavior in (b) is a geometrical consequence as the angle approaches 90° at infinite heights. Although this almost linear, coarse tuning approach via loop numbers is sufficient for AFM nanoprobes, some might ask for more precise fine control. In this case, DT tuning is very powerful, as shown in Figure 3c for 430 pA, sp = 1 nm and fp = 42 nm for *l* = 1. As evidenced by a DT sweep from 10 ms to 25 ms, this approach enables precise height control (dashed line) from 2.5 µm to 5 µm (shaded blue) but in a gradual manner compared to the discrete variation for the repetition of loops (b). In conjecture, very precise control of achievable side wall angles and thus overall heights is enabled. For our situation, we decided on a constant DT of 12 ms for further experiments to keep the loop numbers and therefore the total number of patterning points low.

Finally, we focused on the apex itself, as it is of essential relevance for AFM nanoprobes. Figure 2 and Figure 3 show spiky features at the topmost termination points. The common element of these tips is that the pattern terminated at a single, central patterning point. If this last point was exposed to longer DTs in range of seconds, a distinctive pillar would form, as shown in Figure 4a on the top of a half-HC (red top part). However, when the last patterning point is terminated with the same DT as for the entire HC, the situation becomes slightly unpredictable and results in small and fragile structures on top of the HCs, as indicated in (b) by the red arrow. Such features are highly unwanted for AFM if not controllable. Therefore, the patterning strategy was changed and did not entirely close the rings towards a single pixel point. 

Figure 4 shows a direct apex comparison for patterns that were fully closed (b) and stopped after reaching an inner radius of 20 nm (c) and 50 nm (d). Whereas the latter led to a small indent/opening (red arrow), the intermediate value provides a very well-shaped apex with reproducible radii of approximately 10 nm and less. The critical value determined here for the inner radius is in a similar range as the specified beam diameter (~17 nm), which might be a generic rule of thumb for replication with other currents/machines. After completing the initial fabrication, the controlled material transfer into pure metals is the next step.

### 3.2. Purification

In this section, we focus on the material transfer of as-grown HCs into all-metal structures to fulfill the functional requirements for CAFM operation. In analogy to the deposition process, the purification procedure also relies on a carefully set-up electron beam, highly controlled beam movement realized via a pattern file and a certain atmosphere inside the microscope. SEM imaging, EDX and electric resistivity measurements, as well as TEM investigations, were used to analyze the purification process and relate the changes in microstructure to changes in physical properties. This section will start with discussing the electron-beam setup and the atmosphere inside the microscope. In the following sections, a series of EDX measurements are shown to elaborate the dynamic behavior of the material composition during purification. Using a micromanipulator, the electric resistivity was evaluated as a measure of the success of purification. TEM investigations help to correlate the changes in microstructure with tuned physical properties and thereby underline the effectiveness and necessity of the purification process.

#### 3.2.1. Beam Setup

To ensure proper purification, the penetration depth of primary electrons, which impinge the as-prepared HC from the top [23], has to be considered. For optimized electron penetration, *E*_0_ must be appropriately chosen to provide in-depth purification, avoiding only partial surface purification. Figure 5 shows an FIB-processed HC with an overlay of simulated electron trajectories in the as-deposited material for 5 keV (a) and 30 keV (b). As evident, high-energy electrons are able to deeply penetrate the material, which ensures proper dissociation of inwardly diffused water molecules to purify the entire HC with high efficiencies [23,24]. Consequently, only 30 keV primary electron energies were used for HC purification in the following experiments. Next, the impact of beam currents (I_0_) was explored through a series of purification experiments. Figure 5c shows an overview of dynamic purification experiments performed at 26 nA (green) and 62 nA (red) by in situ monitoring of two main quantities: (i) the relative HC diameters (solid line, left ordinate) and (ii) the EDX-based peak ratios (symbols, right ordinate). Considering the former, the expected volume loss of about 2/3 [23] should lead to a diameter reduction of about 1/3, assuming a spatially homogenous volume loss in 3D space. However, such an assumption would imply a free contraction of the base, which is not necessarily given in the case of tight HC–substrate bonding. Although slightly higher in real experiments, both diameter trends reveal a saturating behavior, resulting in a qualitative but valid evaluation quantity for purification. For in situ EDX analyses, spectra were continuously acquired with a 60 s basis for the full purification period and analyzed using the previously introduced semiquantitative approach, whereby the ratio of integrated peak areas of the C-Kα peak (180–360 eV) and Pt-Lα (9.26–9.54 keV) are calculated. In previous studies [12,25,28], it was demonstrated that the saturation of this ratio informs the dynamic purification saturation, which is represented by the symbols in Figure 5c. The data shown correspond to the same max. purification time of 45 min, which was later recalculated into e-beam doses, which explains the different plot lengths for the two currents.

Aside of the saturating tendencies, the qualitative agreement between EDX and diameters is evident. Interestingly, low currents seem to allow for more efficient purification (faster decay) with slightly lower peak ratios and smaller base diameters. We attribute this observation to the fact that lower currents enable a gentler material transfer (purity and volumetric contraction), whereas high currents can lead to H_2_O depletion effects and therefore taking slightly longer to achieve full purification. The SEM side-view images above the plot in Figure 5c depict the dose-dependent morphological evolution for 26 nA (framed green), which reveals a surface rippling effect with increasing doses. Although gradual, the effect becomes stronger as the purification itself is widely completed, which we attribute to increasing heating effects, leading to grain coalescence, as evident in the SEM image at high doses of 5.5 × 10^3^ C·cm^−2^. In many cases, such high doses also led to a thin surface contamination layer, as evident in the same inset. As such rippling also affects the apex region, intermediate doses of approximately 2.5 × 10^3^ C·cm^−2^ were applied for further experiments to provide sufficiently smooth surfaces and sharp apices for reliable AFM operation. As discussed later, that selection is also motivated by electric results, providing high conductivity at the same doses. In comparison, purification at 62 nA led, in most cases, to implications of the apex regions, as shown for 5.5 × 10^3^ C·cm^−2^ by the red-framed SEM image on top. Consequently, the previously mentioned beam currents of 26 nA at intermediate doses were applied for further experiments to provide well-shaped apices and smooth surfaces, both of which are required for high-quality AFM operation.

#### 3.2.2. H_2_O Pressure

Considering the aforementioned possibility of depleting H2O concentrations during purification at high beam currents, the partial pressure was systematically varied, together with I_0_, to explore possible improvements. Figure 6 summarizes the findings from a morphological point of view concerning relative height (a) and diameter (b) shrinkages. As evident in both plots, the highest beam currents of 111 nA (blue) led to the lowest volumetric decrease without clear pressure dependencies. In contrast, the lowest beam current of 26 nA (green) led to the highest volume losses in both dimensions, with a slight increase in diameter losses at higher pressures (b). Figure 6c shows side-view SEM images after purification at the lowest beam currents with increasing pressures (from left to right) and identical e-beam doses. Figure 6d shows the same experiments with the highest beam currents of 111 nA and increasing doses—this time from right to left to provide a direct comparability of low- and high-current results at the highest pressures in the center (yellow arrow). In agreement with the quantitative data in (a,b), there are only weak dependencies on the pressure, supporting the hypothesis that very high beam currents quickly deplete the H_2_O concentrations, which in turn reduce the efficiencies even for the highest applicable water pressures in our system. However, for the lowest beam currents, the entire process seems to be more efficient, which is also beneficial concerning the stable apex shapes, as discussed before. Although higher partial pressures seem to slightly improve the situation (b), e-beam focusing is more complicated for *p* > 80 Pa, which is why further experiments were performed with 26 nA and 80 Pa.

#### 3.2.3. Purification Pattern

Finally, DT and PoP variations during purification are discussed to explore possible implications for completeness. According to our fabrication patterning strategy discussed above, symmetric pixel distances for fast and slow pitches were chosen (hereafter referred to as PoP for simplicity). Figure 7a summarizes height (squares) and diameter losses (circles) for two different DTs; overall doses were kept constant by an adaption of patterning loops. As evident, height losses first increased up to the PoP range of 30–40 nm, followed by another weak decrease. In contrast, diameter losses were found to be widely constant up to the same PoP range, after which a slight decrease was observed. As evident in the same plot, longer DTs (red) seem to be preferable, as they lead to stronger volume losses at same doses compared to short DTs (blue). Complementary SEM images of HCs at selected point pitches for DT = 100 µs are shown in (b–d). The reported PoP threshold can be interpreted as a situation in which the beam overlap does not fully purify proximity regions, leading to reduced efficiencies during the material transfer. The fact that the reported range is about twice the beam diameter for the given current (~17 nm) might be attributed to the skirt effect in environmental SEMs, which broadens the beam size due to gas-phase scattering effects [39]. Therefore, PoPs and DTs of 35 nm and 100 µs, respectively, were chosen for all further experiments, which, together with the selection of 80 Pa and beam currents around 26 nA, complete the HC purification studies.

## 4. Characterization

### 4.1. Strucural, Electrical and Mechanical Investigations

The large volume loss upon purification naturally raises the question as to how apices are affected by the purification procedure. Accordingly, finished HCs were subjected to transmission electron microscopy studies without any further preparation to allow for a native view of final morphologies. Figure 8 shows an in-scale comparison of an as-deposited HC (a) with a fully purified HC (b) via scanning TEM high-angle annular dark-field (STEM-HAADF) imaging. After fabrication, the apex reveals an FEBID-typical metal-matrix composition with small Pt grains of 2–4 nm [11] and apex radii slightly larger than 10 nm (a). After optimized purification, the apex radii are maintained and often even smaller, whereas Pt grains coalesce with average diameters of 6–10 nm (b), and the overall shape is also maintained. When capturing high-aspect-ratio apex morphologies, the actual HCs can be terminated by short additional pillars, as shown in Figure 4a. The effect of the purification procedure is shown in (c,d), which reveals a remarkable morphological stability, together with the anticipated volumetric reduction, again leading to sharp apices. Such modifications illustrate the highly useful flexibility of 3D-FEBID, whereby the morphology can be tuned according to the required demands, such as high-aspect tips, as shown here, or other designs.

Next, we focused on an electrically basic characterization, which was achieved via SEM micromanipulators for fast conductivity screening as a function of the applied purification dose. To that end, HCs were prepared directly on the Au electrodes of self-sensing cantilevers and further subjected to purification at varying doses. For characterization, a micromanipulator was preshaped and carefully cleaned via FIB. Only low-dose SEM navigation was employed to minimize any unwanted HC surface contamination, which could convolute the resistance measurements. The results are summarized in Figure 9a, where the reference resistance of the Au electrode is indicated by blue measurements point on the left. The red symbols indicate the final electrical results, and the error bar corresponds to the distribution of multiple measurements on the same and identically fabricated HCs. After starting at resistances in the MΩ range for as-deposited HCs, the decreasing and eventually saturating resistance is clearly evident, proving the successful material transfer into electrically conductive materials, as intended. The typically achieved resistances around 1.5 kΩ are slightly higher than reference values at Au electrodes (~1 kΩ) and can therefore be considered as the relevant HC resistance, including contact resistances to the Au electrodes. The finding of ~1 kΩ for the sole electrode measurements is attributed to the long measurement paths and the contact resistances of the rather simple setup. Related doses for saturation are found in the same range as for the volumetric/EDX analyses shown in Figure 5c, which validates the aforementioned purification parameters for ideal material transfer protocols (~2.5 × 10^3^ C·cm^−2^).

After those measurements, purified HCs were subjected to a basic qualitative mechanical characterization. The micromanipulator front parts were adopted accordingly via FIB to compress the pillars for qualitative evaluation of plastic deformation and/or fractures. In particular, the pillars were subjected to cyclic compression with vertical displacements up to 500 nm on very stiff self-sensing cantilevers with spring constants of approximately 140 N·m^−1^. Figure 9 shows a representative SEM series of the situation before (b), during (c–e) and after compression (f); the left, green-shaded part is taken for direct comparison from (b). The noisy appearance is a consequence of the aforementioned low-dose SEM imaging to minimize any e-beam-related implication. As evident, the overall shape is maintained without deformation and/or fracture, and the apex itself often suffered in situations in which the micromanipulator rapidly slipped sideways due to imperfect contact geometries (see red arrow). Finally, tip fracture was provoked by exposing HCs to high lateral forces exerted by the micromanipulator. Figure 9g shows a typical fracture result, exhibiting sudden behavior without plastic deformation. Most importantly, however, delamination at the bottom of the HCs was never observed, which clearly indicates strong bonding to the Au layer, which is essential for reliable AFM operation. Figure 9h shows a high-resolution image of a fractured HC in which the nanogranular composition across the entire HC structure is clearly evident, further underlying the ideal material transfer. Therefore, it can be concluded that morphological, structural, mechanical and functional aspects of the proposed fabrication route are highly promising for next steps, which focus on AFM operation.

### 4.2. AFM Operation

In this final step, purified HCs were subjected to AFM operation for final performance evaluation. To that end, two different AFM platforms were used: FastScanBio from Brucker and the AFSEM™ from GETEC Microscopy. The former was used for classical AFM imaging, whereby standard cantilevers were truncated via FIB processing, followed by fabrication and purification of HCs. The latter system was designed for integration in highly space-confined SEMs, FIBs or dual-beam microscopes via the application of a piezoresistive self-sensing cantilever to eliminate optical laser/detection components. Here, HCs were fabricated directly onto the prestructured Au electrodes of the self-sensing cantilever, allow for correlative in situ electrostatic force microscopy (EFM) and conductive AFM (CAFM) measurements in high-vacuum environments of SEMs, FIBs or SEM-FIB dual-beam microscopes. The fact that HC fabrication must be performed on prefinished and non-flat elements clearly demands an additive, direct-write technology, such as 3D-FEBID, as alternative methods are barely applicable on such surfaces. The combination of 3D capabilities and nanoscale precision, clearly demonstrates both the motivation of this study in particular and the advantages of 3D-FEBID in general.

#### 4.2.1. AFM Height

Even during first tests on standardized reference samples and, in particular, on Au nanoparticle specimens, the superior lateral resolution compared to traditional EFM/CAFM tips became immediately evident. This stems from the fact that our HCs do not employ any coatings, as they have an all-metal character, including sharp apices in the 10 nm region and below, as originally intended. For advanced demonstration, a Si-SiO2 sample with an additional ~70 nm thick Au layer produced via physical vapor deposition was used. These samples were then structured via FIB to open the Au layer in a variety of different shapes for further AFM comparisons. Figure 10 shows 5 µm wide AFM height images of such structures, which were acquired in tapping mode using a standard AFM tip ((a), OTESPA by Bruker), our FEBID-based HC (b) and a PtIr-coated tip ((c), CONTV-PT by Bruker) intended for electric AFM operation. As evident, our HC nanoprobes (b) provide an image quality comparable to that of the uncoated reference tip in (a), revealing all details of the same sample. In contrast, the metallically coated EFM tip (c) provides much less lateral resolution and higher noise compared to the relevant HC tips (b). Figure 10d shows cross-sectional profiles from the indicated lines in (a–c) using the same color code. As evident, HCs (blue) provide practically identical morphological information about side wall slopes when compared to the standard reference tip (black). However, PtIr-coated tips (red) are subject to natural limitations and do not allow for reliable trench imaging due to their additional functional coating, which strongly increases the conical diameter, as well as the apex radii. The green-shaded region of two trenches is shown in more detail in (e), which confirms the superior morphological resolution performance of HC nanoprobes compared to traditionally coated tips for electric operation.

#### 4.2.2. Electrostatic Force Microscopy

EFM is a multipass technique that first acquires the topography of a single line, then lifts the tip to a certain height (h_L_) and scans across the profile once again while monitoring the phase shift. When h_L_ is set to higher than the typical van der Waals interaction range of a few nanometers, the phase shift is dominated by the electrostatic far field, providing qualitative information about the surface potential and possible polarization contributions [29]. Figure 11 shows 3D topography images from a Si-SiO_2_-Au (~70 nm) multilayer sample, which was incompletely milled via FIB to obtain some remaining Au-on-SiO_2_ islands for EFM testing. Whereas (a) shows the tapping height acquired with a fully purified HC, (b) shows the corresponding tapping-phase image as a colored overlay, and (c) includes the EFM phase shift as an overlay, which was acquired at h_L_ = 50 nm and without any bias voltage applied to the tip and/or the sample. The first detail is the very different information between the tapping (b) and EFM phases (c), as representatively indicated by arrows. Especially in narrow trenches, the latter provides strong contrast and informs about different surface potentials as a qualitative marker for different materials. Second, the correlation of EFM contrast with the underlying morphology reveals excellent lateral correlation, which underlines the achievable lateral resolution with such HC nanoprobes. In a different study, the same HCs were used for an AFSEM™-based, correlated SEM-AFM-EFM study on barium titanate samples, where the formation of potential barriers as a function of chemical composition was successfully characterized [40]. Both studies clearly demonstrated the high performance of the proposed HC concept, which benefits, in particular, from the all-metal, sub-10 nm apex as discussed above.

#### 4.2.3. Conductive AFM

Finally, fully purified HCs were used for CAFM measurements with different samples to evaluate the overall performance. The sample in Figure 12a–c is a thermally grown Cu surface further modified by graphene multilayers via chemical vapor deposition and decorated by non-conducting Sb particles. The fibrillar features in the AFM height image (a) correspond to Cu features and reveal lateral widths down to about 40 nm (yellow arrows), which are resolved by the HCs due to the sharp apices. The corresponding CAFM image in (b) reveals several aspects: first, the graphene multilayers are evidenced by the sharp edges due to discrete, layer-related conductivity variations, as indicated by the white arrows, which cannot be clearly identified in the corresponding height image in (a). Second, a varying conductivity across Cu fibers (yellow arrows) is evident in the CAFM map (b), whereas a representative ∆I cross section (taken from the black, dashed line (b)) is shown in (c), which confirms both low-current sensitivity and lateral nanoscale resolution. Sb particles are confirmed as non-conductive, as expected (black regions in (b)), and additional features with an intermediate conductivity were found, displayed as greenish in (b), the widths of which are well below 40 nm in both height and conductivity maps. These results indicate the high performance of the proposed HC concept, which clearly benefits from both the all-metal character and the sharp apices for proper nanoscale characterization. In another experiment, a conductive Cu sample, which was partially coated with a semiconducting MoS2 layer, was investigated via CAFM with the aim of identifying incompletely covered regions, together with electric basic spectroscopy. Figure 12d shows a 3D height representation with the conductivity map as a colored overlay. To study electric properties in more detail, stationary I/U spectroscopy was performed in selected areas identified by the conductivity map. The highly conductive regions, appearing as white in (d), revealed an ohmic behavior, as shown in (e), which relates to the incompletely coated Cu region. In contrast, the surrounding regions, which appear as blueish in (d), reveal a semiconducting behavior, as expected for the MoS2 layer, as indicated by a representative I/U curve in (f). Therefore, the suitability of the proposed FEBID-based hollow cones for CAFM operation with respect to lateral resolution and current sensitivity is successfully demonstrated.

## 5. Conclusions

Herein, we introduced an FEBID-based 3D nanoprobe concept for application as an electrically conductive AFM tip. The basic design is a PtC_X_ hollow cone (HC), which is transferred into pure Pt materials by our well-proven e-beam-assisted purification process in low-pressure H_2_O environments. The results are 3D nanoprobes that are mechanically robust and electrically conductive, with apex radii in the 10 nm range and less on a regular basis. Compared to alternative tips for CAFM, EFM or KFM, which are mostly coated by additional metallic layers, HCs have an all-metal character. Therefore, any risk of delamination is eliminated, which, in turn, increases measurement reliability. Furthermore, the much sharper apices, which can be achieved by FEBID, provide superior lateral resolution as a key element for AFM-based electric nanoscale characterization. In addition to this specific application, which is commercially available, this study reflects the relevance of 3D-FEBIDs as additive, direct-write manufacturing technology for situations in which traditional nanofabrication approaches are very challenging or even impossible. Whereas precision, predictability and reproducibility have matured, the single-beam character remains, resulting in a lower throughput than for mass production tools. In addition to this main limitation, which could be compensated by, e.g., multibeam instrumentation, there are further future challenges, such as new precursor materials or high-purity precursors, as comprehensively discussed in a recent review article [5]. Nevertheless, the application discussed herein demonstrates the suitability of 3D-FEBID for industrially relevant application, owing to its flexibility, reliability and overall performance at the lowest nanoscale, especially when direct-write fabrication is not optional but indispensably required.

## Figures and Tables

**Figure 1 nanomaterials-12-04477-f001:**
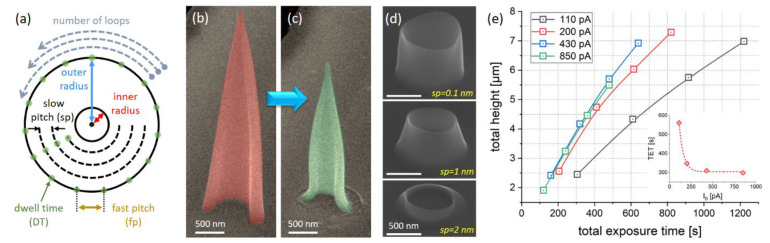
(**a**) Summary of patterning details for hollow-cone (HC) fabrication as discussed in the main text. (**b**) Titled SEM image of a half-HC after fabrication to prove the hollow nature. (**c**) Result of full purification by an in-scale comparison. (**d**) Impact of *sp* variation from 0.1 nm over 1 nm to 2 nm, whereas other parameters were kept the same. (**e**) Total HC heights as a function of total exposure times (TETs); the inset shows the required TET for 4 µm tall HCs, showing saturation for beam currents higher than 430 pA.

**Figure 2 nanomaterials-12-04477-f002:**
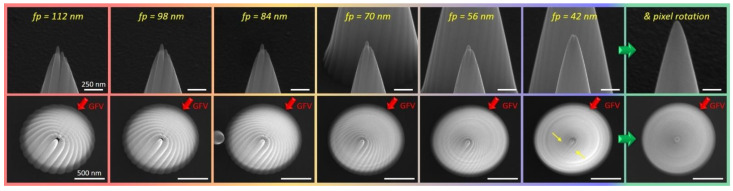
Implications of decreasing fast point pitches fp, shown by tilted (**top row**) and top-view SEM images (**bottom row**), together with fp values and the gas flux vector (GFV) orientation (red arrows). The images on the right depict the implications of cyclically rotated patterns at fp = 42 nm.

**Figure 3 nanomaterials-12-04477-f003:**
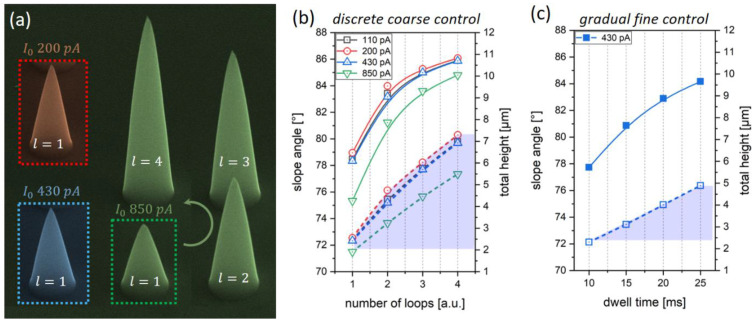
Coarse and fine tuning of total heights. (**a**) Collage of HCs fabricated at increasing currents by a single loop (*l* = 1), with DT, sp and relative fp (beam overlap) values kept constant. (**b**,**c**) Average slope angles (solid lines, left ordinates) and total heights (dashed lines, right ordinates) for coarse control relative to the number of loops and fine control via adjusted DTs, respectively.

**Figure 4 nanomaterials-12-04477-f004:**
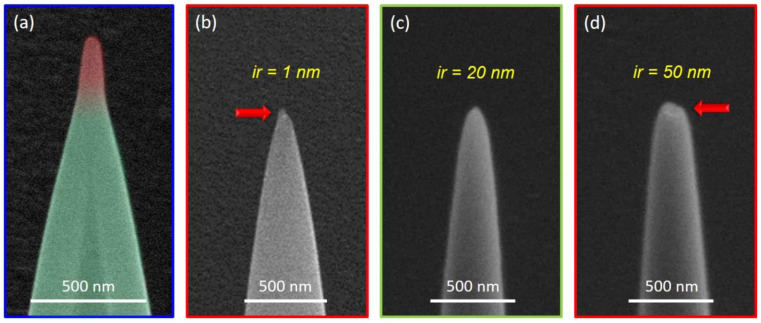
Patterning implications on the tip apex. (**a**) A half-HC that was terminated with a longer DT for the final pixel point, leading to an additional, tunable pillar, as indicated in red. (**b**–**d**) Regular HCs, where the inner radius (ir) of the last patterning ring was gradually increased from 1 nm to 20 nm and 50 nm. Fully closed patterning mostly led to small, unwanted features (**b**), and excessive ir led to an indent feature (**d**), both indicated by red arrows. For appropriate values in a similar range to that of the beam diameter, the structure is reproducibly closed (**c**).

**Figure 5 nanomaterials-12-04477-f005:**
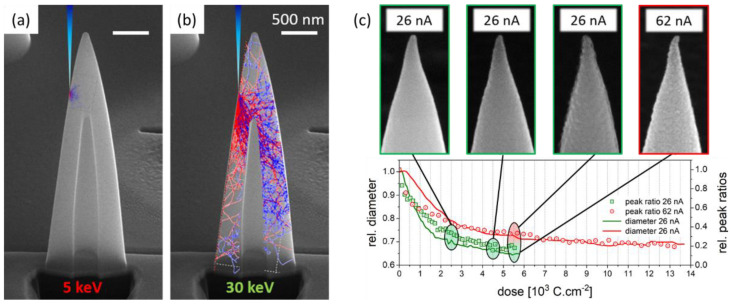
Impact of purification energies, beam currents and total doses. (**a**) FIB-processed HC after fabrication with a colored overlay of electron trajectories at E_0_ = 5 keV in direct comparison to 30 keV (**b**), which reveals the much-improved electron penetration for higher energies as required for efficient purification. The main graph in (**c**) shows the in situ evolution of relative diameters for two different I_0_ (see legend), indicated by solid lines, as a function of applied doses. The symbols represent EDX-based areal ratios of the carbon Kα peak (180–360 eV) and platinum Lα peak (9.260–9.540 eV). The SEM insets on top show the HC top region taken at different stages of purification at 26 nA (green) and 62 nA (red), revealing an unwanted rippling effect for higher doses.

**Figure 6 nanomaterials-12-04477-f006:**
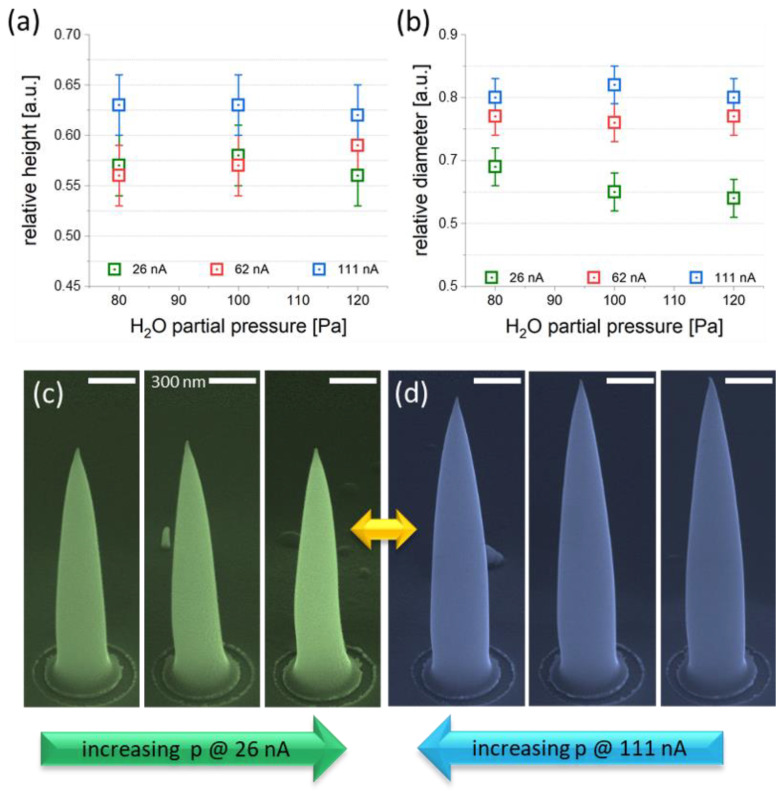
Height (**a**) and diameter shrinkages (**b**) as a function of beam currents and H_2_O partial pressures (see legends). (**c**,**d**) HCs after the same dose at 26 nA (green) and 111 nA (blue), respectively. In agreement with (**a**,**b**), higher volumetric losses are achieved by lower currents, with a minor pressure dependency.

**Figure 7 nanomaterials-12-04477-f007:**
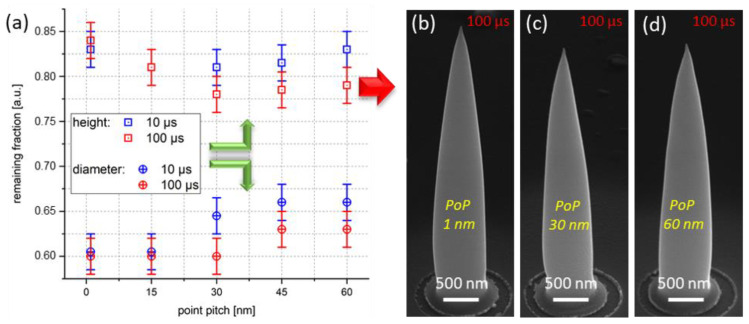
Impact of PoPs and DTs during purification. (**a**) Height (squares) and diameter losses (circles) for two different DTs (see legend) as a function of PoPs, which were selected as symmetric concerning for fast and slow pitches. As evident, a PoP threshold range of approximately 30–40 nm was observed, after which the volumetric shrink decreased again. (**b**–**d**) A series of HCs after purification via different PoPs; DTs and total doses were kept constant.

**Figure 8 nanomaterials-12-04477-f008:**
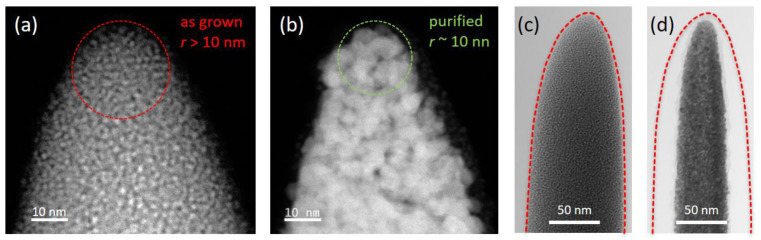
(**a**) STEM-HAADF image of am HC apex region immediately after deposition, revealing a typical metal-matrix composition with a radius of slightly more than 10 nm. (**b**) A typical HC after full purification, with larger grain sizes and a maintained overall shape with an apex radius in the same range or even slightly below. (**c**,**d**) Optional top pillar (see Figure 4a) before and after purification, respectively, revealing remarkable shape stability with maintained apex radius, e.g., for high-aspect-ratio applications.

**Figure 9 nanomaterials-12-04477-f009:**
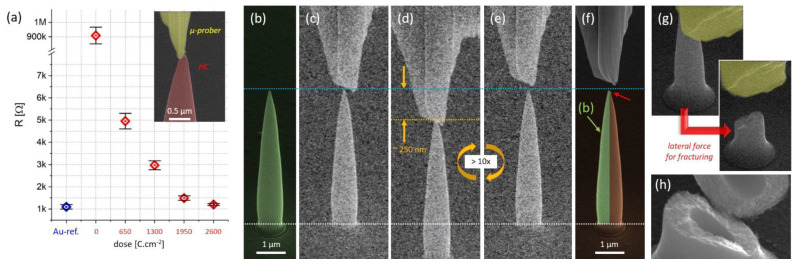
Electric and mechanic basic characterization. (**a**) Electric HC resistivities as a function of purification dose, together with reference measurements on the surrounding Au electrodes (blue). As evident, there is a saturation tendency around 2.5 × 10^3^ C·cm^−2^, in agreement with in situ studies summarized in Figure 5c. The inset depicts the measurement setup by means of an FIB preshaped micromanipulator (yellow) and the actual HC of relevance (red). For basic mechanical evaluation, fully purified HCs (**b**) were compressed in a cyclic manner, as shown in (**c**–**e**). (**f**) Direct comparison of a cyclically stressed HC with the original shape (shaded green and taken from (**b**)), which reveals no morphological implications, with the exception of a slightly damaged apex (red arrow) due to contact problems with the micromanipulator during compression cycles. (**g**) A fractured HC. (**h**) High-resolution SEM image of the fractured region, in which the nanogranular composition across the entire cross section becomes clearly evident.

**Figure 10 nanomaterials-12-04477-f010:**
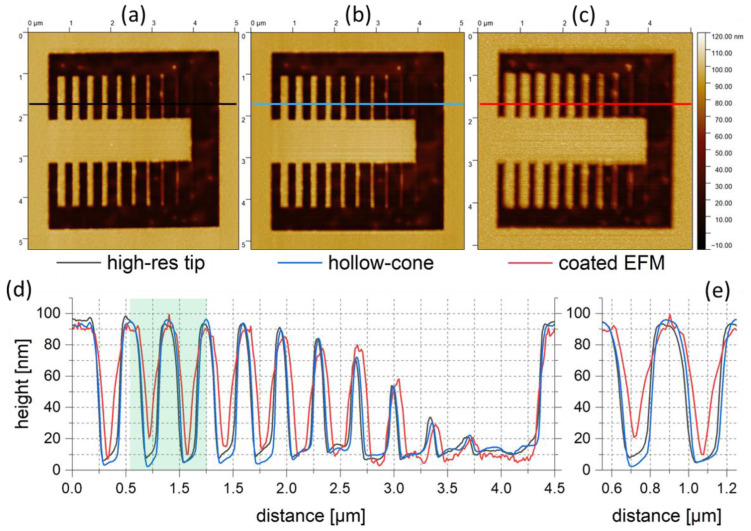
HC topography performance. For morphological benchmarking, an FIB-processed Si-SiO_2_-Au (~70 nm) multilayer sample was subjected to tapping-mode measurements with different tips. (**a**) Height image of a typical uncoated AFM nanoprobe. (**b**) Result for fully purified HCs in direct comparison with a PtIr-coated AFM tip for electric AFM measurements. (**c**) Direct comparison of individual features, clearly demonstrating that the topography performance of HCs is practically identical to that of the standard tip (**a**), with electric conductivity onboard. (**d**) Three cross-sectional profiles taken from the lines indicated in (**a**–**c**) with the same color code. (**e**) The green-shaded part in (**d**) presented with more detail, demonstrating the advantages of FEBID-based HCs (blue) compared to commercially available CAFM/EFM tips (red).

**Figure 11 nanomaterials-12-04477-f011:**
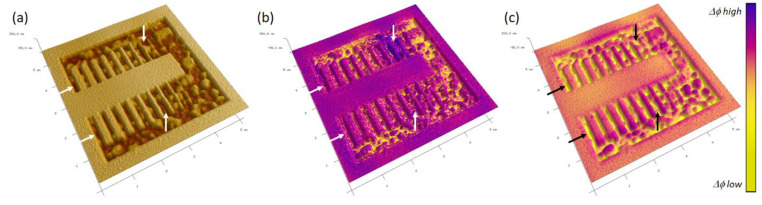
EFM performance. (**a**) A 3D height image of an FIB-processed Si-SiO_2_-Au (~70 nm) multilayer, which was incompletely milled to obtain remaining Au islands. (**b**,**c**) The same height information with a colored overlay stemming from the tapping and EFM phases in lift mode. The distinct information in narrow trenches (arrows) is clearly evident, confirming the appropriate EFM operation, even under unbiased conditions.

**Figure 12 nanomaterials-12-04477-f012:**
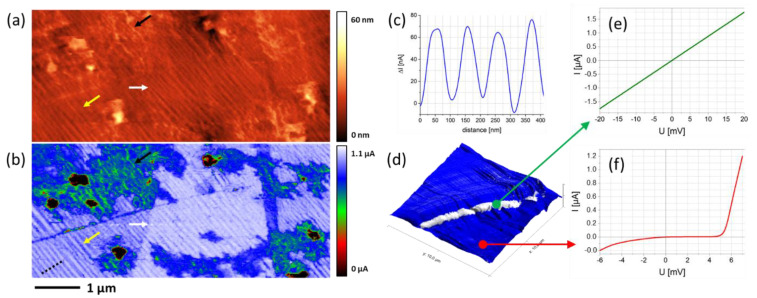
CAFM Performance. (**a**–**c**) A thermally grown Cu sample further modified by CVD-based graphene layers and decorated by Sb particles. The AFM height image (**a**) reveals fibrillar Cu features with well-resolved lateral widths down to about 40 nm (yellow arrows). The CAFM map (**b**) reveals graphene multilayers as evidenced by both sharp edges and discrete jumps in its conductivity (white arrows). A selected ∆I cross-sectional profile is shown in (**c**), taken from the black dashed line in the bottom-left corner of (**b**), which confirms the electric sensitivity, as well as the lateral resolution capabilities. (**d**–**f**) Results from a Cu surface that was incompletely coated by an MoS_2_ layer. (**d**) A 3D topography image with the I-map as a colored overlay. (**e**,**f**) I/U measurements under static conditions taken from the two very different conductivity regions, as representatively indicated in (**d**). Whereas the former reveals ohmic behavior related to the Cu surface, the latter shows semiconducting behavior, as expected for MoS_2_-materials.

## Data Availability

Not applicable.

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
