# Peer review of "3D Nanoprinting of All-Metal Nanoprobes for Electric AFM Modes"

_nanomaterials, 2022, doi:10.3390/nano12244477_

Round 1

Reviewer 1 Report

-       The papers deals with the topic “3D Nanoprinted All-Metal Nanoprobes for Electric AFM Modes”; the topic is up-to-date. The applied 3D nanoprinting is available for producing all-metal nano-probes for atomic force microscopy (AFM) based electrical operation modes. The method is the so-called FEBID.

-       The study uses design aspects to motivate the introduced HC (hollow-cone) architecture, followed by detailed fabrication characterization to identify and optimize FEBID process parameters.

-       The microstructure of final HCs was analyzed via STEM-HAADF, while electrical and mechanical properties were investigated in situ using micromanipulators. The AFM/EFM/CAFM measurements were executed in comparison to non-functional, high-resolution tips and commercially available electric probes.

-       The authors stated that the here introduced all-metal HCs ensure the resolution capabilities of the former, while having the electric conductivity of the latter onboard thereby combining both assets in one design.

-       The applied method is clear, the introduced and fulfiled-executed measurements are satisfactory.

-       The result and the discussion is unequivocally state the relevant outcomes and conclusions.

-       The prepared and pasted figures are clearly understoodable, the resolutions are high enough. The connecting explanations are adequate.

-       The applied references are relevant and up-to-date, however, the used reference-citation style does not fit to the MDPI journals’ requirement. Please modify and revise it!

-       The format of the manuscript (i.e., formatting) does not suit in many locations the MDPI journals’ template. Please correct them!

-       The reviewer recommends a Nomenclature based on the applied parameters and their meaning (i.e., short explanations), as well as the related units in SI. The Nomenclature should be in the last part of the paper.

-       In the abstract, there are many abbreviations without explanation (giving the meaning), i.e., e.g., FEBID, STEM-HAADF, etc. Please supplement them and revise it!

-       The composition of the manuscript needs revision. In scientific paper the use of “we” and the connecting active mode are not allowed. Please modify the sentences to full passive mode, or in the case of applying active mode: exchange the “we” to “the authors”.

-       There are many units in the text with number, however, they are with Italic style. Why?

-       The structure of the manuscript must be revised because the reader can’t decide which are the Sections and Subsections, as well.

Author Response

Dear editor, dear reviewer!

Please find our response in the attachment!

Yours sincerely, 

Harald Plank

Reviewer 2 Report

This study starts with design aspects to motivate the here introduced HC architecture, followed by detailed fabrication characterization to identify and optimize FEBID process parameters. To arrive at desired material properties, e-beam assisted purification in low-pressure water atmospheres at room temperature is applied, which enables the removal of carbon impurities from as-deposited structures. The article deserves publication in Nanomaterials after addressing the following minor comments:

1. If the journal style allow, it will be better to list a table of notations.

2. Some of the figures needs to be rendered with a better resolution.

3. The purification process needs to be elaborated more in the discussion.

4. Conclusion section is required.

Author Response

(The authors gave the same response as above.)

Reviewer 3 Report

This work starts with design aspects to motivate the here-introduced HC architecture, followed by detailed fabrication characterization to identify and optimize FEBID process parameters. The paper's contribution to existing knowledge in this research field is well justified. The paper needs to contribute more, and the following points can improve the manuscript.

1.     Enhance the introduction to show the motivation for this work.

2.     A comparative study can be added to a related work section in table form.

3.     Figures 1, 2, …, and 12 captions should be simplified. The details can be shifted to the main text.

4.     The manuscript organization should be improved. Sections and subsections need to be clearer.

5.     A list of abbreviations can be added in the introduction section or at the end of the manuscript.

6.     There should be some discussion on the limitations of the methods presented in a separate section.

7.     Improve the English of the work. There are too many problems with paper typesetting.

8.     Change the “Conclusion” section title to “conclusion and future directions” and add more discussion and future directions to the research.

9.     The paper is unsuitable for acceptance in its current form. The article needs rewriting to address the comments mentioned above. 

Author Response

(The authors gave the same response as above.)

Reviewer 4 Report

The study is interesting and the reviewer believes that the manuscript can be published in this journal. 

There are some points that need to be improved:

- Regarding the H2O partial pressures the temperature was kept constant? The temperature could affect the height and diameter shrinkages.

- Regarding the impact of purification energies the authors need to consider the temperature.

Author Response

(The authors gave the same response as above.)

Round 2

Reviewer 3 Report

The authors have addressed most of my concerns. The paper can be accepted.